# The Similar and Distinct Roles of Satellite Glial Cells and Spinal Astrocytes in Neuropathic Pain

**DOI:** 10.3390/cells12060965

**Published:** 2023-03-22

**Authors:** Aidan McGinnis, Ru-Rong Ji

**Affiliations:** 1Center for Translational Pain Medicine, Department of Anesthesiology, Duke University Medical Center, Durham, NC 27710, USA; 2Department of Cell Biology, Duke University Medical Center, Durham, NC 27710, USA; 3Department of Neurobiology, Duke University Medical Center, Durham, NC 27710, USA

**Keywords:** dorsal root ganglia, fatty acids, gliopathy, nerve injury, spinal cord

## Abstract

Preclinical studies have identified glial cells as pivotal players in the genesis and maintenance of neuropathic pain after nerve injury associated with diabetes, chemotherapy, major surgeries, and virus infections. Satellite glial cells (SGCs) in the dorsal root and trigeminal ganglia of the peripheral nervous system (PNS) and astrocytes in the central nervous system (CNS) express similar molecular markers and are protective under physiological conditions. They also serve similar functions in the genesis and maintenance of neuropathic pain, downregulating some of their homeostatic functions and driving pro-inflammatory neuro-glial interactions in the PNS and CNS, i.e., “gliopathy”. However, the role of SGCs in neuropathic pain is not simply as “peripheral astrocytes”. We delineate how these peripheral and central glia participate in neuropathic pain by producing different mediators, engaging different parts of neurons, and becoming active at different stages following nerve injury. Finally, we highlight the recent findings that SGCs are enriched with proteins related to fatty acid metabolism and signaling such as Apo-E, FABP7, and LPAR1. Targeting SGCs and astrocytes may lead to novel therapeutics for the treatment of neuropathic pain.

## 1. Introduction

Neuropathic pain is a widespread and crippling disability which impacts quality of life, often severely [1]. The international association for the study of pain (IASP) defines this condition as pain resulting from damage to the somatosensory nervous system [2]. Unfortunately, neuropathic pain is often unresponsive to modern therapies, and opioids are still commonly prescribed to refractory patients despite the risks of addiction and respiratory suppression [3,4].

Glial cells are strongly implicated in the pathogenesis of neuropathic pain [5,6,7]. Glia are diverse; the spinal cord is populated by microglia, astrocytes, and oligodendrocytes, while the peripheral nervous system contains Schwann cells and satellite glial cells, as well as enteric glia. Oligodendrocytes and Schwann cells are highly abundant and are responsible for myelin production in the CNS and PNS, respectively, among a growing list of other functions. Schwann cells are emerging as important cells in neuropathic pain pathogenesis, and although the nature of their role in neuropathic pain remains poorly understood [8], it is likely that more is known about them in this context than about oligodendrocytes [9,10]. Microglia are the central nervous system (CNS) resident macrophages and serve a variety of immune-related functions; their role in neuropathic pain has been comprehensively reviewed elsewhere [11,12].

Satellite glial cells (SGCs) and astrocytes are often compared, and they do share many critical functions [13]. Astrocytes are critical to homeostasis in the CNS, where they provide structural support for neurons, help form synapses, regulate the extra-cellular fluid, participate in neurotransmitter regulation, maintain the blood–brain barrier (BBB), and respond to insults [14]. SGCs are more specialized cells which serve similar supportive functions in the context of the peripheral ganglia and are, therefore, not involved in BBB or synapse formation [13]. The morphologies of these supportive glia are notably distinct (Figure 1). Further, while they respond to nerve injury in many of the same ways, these responses are not totally congruent and occur on different timescales, with SGCs in affected DRG becoming active very rapidly (hours to days) [15], while astrocytes in the ipsilateral dorsal horn display phenotypic shifts after days to weeks [16].

## 2. SGCs and Astrocytes in Homeostasis

### 2.1. Location

SGCs are located throughout the ganglia of the PNS, where they play supportive and protective roles for a variety of peripheral neurons. Two major categories of ganglia exist: sensory ganglia house primary sensory afferents, while autonomic ganglia are where the soma of efferent autonomic neurons reside. The dorsal root ganglia (DRG) contain the soma of all sensory neurons which innervate the body below the neck. Above the neck, a series of ganglia corresponding to the facial nerves handles a swath of functions, some sensory and others autonomic. The trigeminal ganglion (TG) is the best studied of these in the context of pain.

Astrocytes are found throughout the central nervous system (CNS), in both grey matter (protoplasmic astrocytes) and white matter (fibrous astrocytes). DRG neurons project to the spinal cord through the dorsal horn ipsilateral to them, where they synapse with spinal neurons which ultimately relay sensory signals to the brain. The role of astrocytes in neuropathic pain is far from restricted to the spinal cord: impressive work has linked astrocytes in the cortex [17,18,19,20,21], periaqueductal gray, [22,23], and thalamus [24] to pain. Whether this activity is primary to or a consequence of chronic pain is a complex question beyond the purview of this review, which will focus solely on spinal astrocytes.

### 2.2. Development

SGC precursors originate in the neural crest alongside cells which will become Schwann cells, smooth muscle, bone, and the rest of the PNS [25]. A growing body of evidence suggests that SGCs are closely related to Schwann cells and may, in fact, be immature Schwann cells held in their present state by interactions with sensory neurons [26]. Supporting this view, SGCs rapidly change their phenotype in monoculture, becoming less broad and more closely resembling Schwann cells or their precursors, termed sympathoblasts [26,27].

Astrocyte precursors originate from the neural tube, which differentiates into the entire CNS [14]. Thus, although the phenotypes of astrocytes and satellite glial cells are quite similar, both glia are more closely related developmentally to the neurons they support than to one another.

### 2.3. Morphology

The differing structures of SGCs and astrocytes reflect the scale of their interactions with the neurons they support. Understanding these unique neuron–glia relationships at homeostasis is essential to the conceptualization of their dysregulation following nerve injury.

Each peripheral sensory neuron is “sheathed” by SGCs, which surround the entire soma and extend onto the beginning of the axon [28] (Figure 2). This so-called “neuron-glial unit” is itself surrounded by a basal membrane, which then interfaces with the ganglia’s connective tissue [29]. Individual SGCs are slightly irregular in shape and may have a number of small projections branching off of the primary cell body [13]. Neurons appear to extend thin processes outwards and make contact with SGCs [30]. Some have suggested that the neuron-facing surface of SGCs is different in structure than that facing outward, meaning these cells are polar, but this remains an open question [13]. Murine DRG neurons have roughly 4–12 SGCs each, a number which is greater for larger cells and in species with larger neurons [31]. The number of SGCs per human DRG neuron remains to be investigated, but is likely much higher, because human sensory neurons are typically more than twice the diameter of their murine equivalents. Roughly 5% of neuron–glia units include two or even three neurons [32]. A small histology study in rabbits suggested that aged animals may have a decreased number of satellite cells [33].

The distance between SGC and the associated neuron has been estimated to be as small as 20 nm, a comparable distance to neuron–neuron synapses. It is not difficult to imagine that this pseudo-synapse allows for rapid, bidirectional paracrine communication. Indeed, the release of cytokines and ATP has been shown to regulate neuronal function [34]. Further, the small volume of extra-cellular fluid present implies that SGCs likely exert significant control over factors such as ion concentration; a resting potential of between −10 and −20 mV has been measured in what was likely the SGC-sensory neuron cleft in cat DRG [35]. Although sensory neurons are tightly enveloped, the gaps between SGCs are large enough to allow macromolecules to slowly reach the neuron [36]. Mechanisms of crosstalk between DRG neurons have been identified [37], and neuron–neuron “coupling” becomes more evident after nerve injury [38], demonstrating that SGCs do not totally isolate neurons from one another.

Although SGCs have a large surface area to volume ratio, astrocytes far exceed them. They are marked by countless processes that extend outwards in three dimensions, enabling them to interact with multiple neuronal somas, hundreds of dendrites, and >100,000 synapses, although contact with any given neuron is relatively limited [39]. Notably, a single human astrocyte domain has been estimated to contact up to 2 million synapses, underscoring their important role in human brain function. Astrocytes are very abundant in the CNS, accounting for between one and two out of every five cells, but the CNS may possess fewer astrocytes than it does neurons [40], in stark contrast to the ratio between SGCs and neurons in the peripheral ganglia.

Both SGCs and astrocytes stand between neurons and the vasculature. While the extent to which SGCs act as a filter between blood and neuron is unclear [13], astrocytes are prominent in their role as the custodians of the blood–brain barrier (BBB). Astrocytic “feet” tightly enwrap all blood vessels in the CNS and prohibit entry of large and/or polar molecules. This makes the delivery of drugs to the CNS especially challenging [41], leaving SGCs as relatively appealing targets for therapeutic intervention.

### 2.4. Diversity

SGCs are canonically thought to be homogenous [34]. A recent scRNAseq study, however, has tentatively separated SGCs into five groups: sensory-ganglion-specific, sympathetic-ganglion-specific, general resident, immune-response related, and those expressing immediate early genes [42]. Still, these differences are thought to be relatively small.

Evidence strongly supports greater diversity in astrocyte populations [13]. Subclasses of astrocytes have been discovered, even within the spinal cord [43]. Single-cell RNA sequencing has identified seven types of astrocytes, two of which—termed “protoplasmic” and “fibrous”—represent canonical astrocytes and have been identified in the mouse spinal cord [44]. Fibrous astrocytes express significantly more *Gfap* than protoplasmic astrocytes, and are more common in white matter [44,45]. Additionally, astrocytes in the superficial dorsal horn, where first-order nociceptive synapses form, appear to have a unique phenotype marked by greater expression of the glutamate/glutamine regulators GLT1 and GS, the gap junction protein Connexin-43 (Cx43), and GFAP, and are more densely packed than astrocytes in more ventral laminae [46]. This population is also marked by the expression of *Hes5* (which encodes a transcription factor) and may serve as a “gate” for mechanical pain through pro-nociceptive α_1A_-adrenoceptors [47]. Astrocytes also contribute to gating pain in the spinal cord under physiological conditions through ATP signaling [48]. Following injury, astrocytes may adopt a pro-inflammatory “A1” phenotype or a pro-regenerative “A2” phenotype [49,50]. Pro-resolution “A3” astrocytes have been theorized to exist, but are not well characterized [49].

### 2.5. Common Non-GFAP Markers

The inability to find genes specific to SGC in the context of the sensory ganglia which are not expressed by astrocytes has hindered work on SGCs using transgenic mice (Figure 3). Both SGCs and astrocytes are enriched with S100 proteins, although they are not the only cell types in these tissues to express them [14]. Sox9 has emerged as a useful marker of mature astrocytes [51]. Single-cell sequencing has identified *Fabp7* as being specific to SGCs in the context of sensory ganglia [52], but it is also expressed by astrocytes.

## 3. SGCs and Astrocytes in Neuropathic Pain

Astrocytes and satellite glial cells rapidly modulate neuronal and synaptic sensitivity and, therefore, have profound impacts on pain processing [34,49]. In naïve rats, activation of spinal astrocytes via optogenetics is sufficient to cause mechanical allodynia and thermal hyperalgesia lasting up to a week [54].

Glial cells respond to inflammation and nerve injury by becoming more reactive—this process is called reactive gliosis. First, these cells respond to injury by upregulating cytokines and chemokines (small cytokines) [55], part of a general trend of increased bidirectional paracrine signaling with the neurons they support, and by producing more GFAP, which is indicative of reactive gliosis. Increases in gap junctions in SGCs are well-documented, and SGCs may also upregulate ATP-gated cation channels and downregulate inwardly rectifying potassium channels (e.g., K_ir_4.1). Glutamate/glutamine equilibrium is also perturbed after nerve injury, but this is only well-characterized in astrocytes [56]. As a result of these dysregulations, extracellular levels of potassium and glutamate are increased [34,49]. Ultimately, the neuron(s) contacted become sensitized, contributing to chronic pain.

SGCs become active immediately upon nerve injury, undergoing gliosis within 2–48 h [15]. Astrogliosis may occur after microgliosis: immediate activation of superficial lamina neurons rapidly activates spinal microglia, but not astrocytes. Within the next week, inflammatory mediators, released particularly by microglia, drive astrogliosis [16,57,58]. It is also possible that nerve injury may directly activate astrocytes in a microglia-independent manner.

### 3.1. GFAP

Glial fibrillary acidic protein (GFAP) is a homopolymeric intermediate filament which serves as the primary filamentous component of astrocyte processes [59]. GFAP is tightly linked to reactive gliosis because of the structure it provides to enlarging processes. Early reports of its detection in the PNS were controversial, and rightly so, as the first generation of antibodies to GFAP often proved incapable of differentiating GFAP and other intermediate filaments [60]. Now, with the advent of GFAP reporter mice and more carefully vetted antibodies, it is widely thought that SGCs in sensory ganglia express low levels of GFAP under homeostatic conditions—much lower than do astrocytes [6,61]. In addition, GFAP has been identified in non-myelinating Schwann cells, which provide critical support to unmyelinated c-fiber afferents and are involved in pain homeostasis [62] and in enteric glia [63,64,65]. There are at least ten known splice variants of GFAP, as well as several isoforms preferentially expressed by some astrocyte populations.

After nerve injury (or inflammation), both SGCs and astrocytes upregulate GFAP [6]. GFAP knockout mice recover faster after spinal nerve ligation-induced allodynia, and intrathecal delivery of GFAP antisense oligonucleotides ameliorates hypersensitivity while downregulating GFAP in DRG in addition to SC [66]. Astrogliosis may also involve the upregulation of vimentin, nestin, and/or synemin, depending on the nature of the insult [67]. These proteins are not as well-characterized as GFAP in the context of neuropathic pain.

Satellite glia have been found to upregulate GFAP in the sensory ganglia of axotomized rats [68], rats with spinal nerve ligation (SNL) [66,69], mice and rats with streptozotocin-induced diabetic neuropathy [70], and in mice with chemotherapy-induced peripheral neuropathy (CIPN) [71,72]. Inflammatory pain models also consistently cause SGC GFAP upregulation [6]. However, reports of negative findings have emerged [73]. RNAseq data are inconsistent, with some datasets finding little change in *Gfap* expression after sciatic nerve injury [27] and another identifying upregulation [74]. Possible explanations include regulation of GFAP at the translational and/or protein degradation levels and alternative splicing producing isoforms not recognized by common monoclonal antibodies. Still, these incongruities have led to some call for SGC researchers to reduce their reliance on this single biomarker [61].

GFAP is highly expressed in astrocytes and is universally accepted as a marker of reactive astrogliosis. Expression at homeostasis is broad but uneven, with astrocytes in the superficial dorsal horn expressing notably more GFAP than those in deeper lamina [9,46]. Sciatic nerve constriction, spinal nerve ligation, spinal cord injury, CIPN, trigeminal nerve injury, and experimental autoimmune encephalitis (a mouse model of multiple sclerosis) have all been shown to cause GFAP upregulation in the murine SC [16,57,75,76,77,78,79,80,81,82]. GFAP also appeared to be more highly expressed in the spinal cord of a human patient with chronic regional pain syndrome (CRPS), and spinal cords from patients with HIV-induced neuropathic pain [83,84]. Intrathecal delivery of astrocyte toxins and inhibitors has shown efficacy in animal models of nerve injury-induced neuropathic pain [79,80].

### 3.2. Reactive Gliosis: Beyond GFAP

Morphological changes in SGCs following nerve injury are a matter of current controversy. Several groups, using BrdU application and subsequent histology, have reported that SGCs proliferate following both nerve injury and inflammatory pain models [85,86,87]. This is theorized to occur in order to provide greater metabolic support as neurons go through the costly process of regeneration. However, others, using flow cytometry and single-cell sequencing, have found no evidence of proliferation [27,74,88]. Jager and colleagues suggested that these BrdU-positive cells are likely to be proliferating macrophages that have invaded the neuron–glia unit [88], a phenomenon which has been previously reported [89]. Supporting this explanation, macrophages were found to have upregulated cell-cycle-related genes at 3 days post-injury, whereas SGCs had downregulated these genes [74]. SGC proliferation could also be dependent on the nature of the injury. Beyond SGC proliferation, others have provided evidence that SGCs can occasionally even undergo neurogenesis and form new small-diameter c-fiber nociceptors, occurring a week to a month after both sciatic nerve injury and CFA induced inflammatory pain [90,91]. At one month post-injury, approximately one to three neurons of a thousand were observed to have arisen from a satellite glia [90]. This idea is not totally foreign, as glial progenitors have been shown to give rise to parasympathetic neurons [92] and SGCs have been shown to have stem-like properties in culture [91]. Still, a provocative and novel finding such as this one must be interpreted cautiously; further work on this topic is likely ongoing and will be of great interest.

In contrast, when astrocytes become activated, they appear to proliferate, and their processes thicken and lengthen [64,93]. Transforming growth factor-activated kinase (TAK1) signaling and a transient upregulation of STAT3 may be primary to these changes [93,94]. Evidence from a study of brain inflammation suggests that microglia, signaling through the complement system, are responsible for triggering a hyperactive, toxic astrocytic phenotype termed “A1”, which is marked by the expression of C3 protein [50,95]. A popular theory is that this transition from microglial activation to astrocyte activation corresponds to the “chronification” of pain [9]. Interestingly, treatment of rats with post-surgical thoracotomy by knocking down spinal ROR2 (receptor tyrosine kinase-like orphan receptors) via intrathecal *Ror2* siRNA could greatly reduce A1 astrocyte polarization and attenuate pain symptomology [96].

How long SGC activation lasts after an acute injury is not well known. Interestingly, in a rat model of spinal stenosis, signs of SGC activation were seen to peak within one week and then decline or resolve while allodynia persisted [15]. Chronic activation of astrocytes, however, is well characterized; activity can linger for many months and may never fully resolve. [49]. This activation has severe consequences. Astrocyte activation can alter synapses in the CNS through both traditional and “gliogenic” long-term potentiation (LTP), causing central sensitization [97,98] (Figure 4). Nerve injury may even cause astrocyte-dependent necroptosis of dorsal horn neurons [75]. Adequate synapse formation during development requires the adhesion molecule hevin (high endothelial venule protein, also called SPARCL1), a glycoprotein which is secreted by astrocytes and enables the interaction of neuraxin (NRX1α) and neuroligin (NL1), bridging pre- and post-synaptic neurons [99]. In the spinal cord, hevin is pro-nociceptive and necessary for generating central sensitization. Hevin is upregulated in astrocytes following nerve injury, potentiating excitatory synaptic transmission. Specifically, hevin rapidly increases the function of NR2B (GluN2B), an NMDA receptor subunit, in spinal cord pain circuits. Hevin KO mice exhibited faster resolution of nerve injury-induced allodynia and, furthermore, re-expressing hevin in astrocytes could cause pain to return [100]. Ipsilateral spinal astrocytes also upregulate the adhesion molecule TSP4 (thrombospondin 4) after nerve injury. Blocking TSP4 notably reduced neuropathic pain symptomology, and intrathecal TSP4 injection is sufficient to cause hypersensitivity [101]. While TSP4 was also found to be upregulated in the DRG after nerve injury [101], its role in the DRG is not well understood.

### 3.3. Gap Junctions

Gap junctions are microscopic channels which connect adjacent cells and allow the direct diffusion of ions and small molecules (e.g., ATP). It takes six protein subunits, called connexins, to form a connexon, and GJs are built when two connexons, embedded in the membranes of separate cells, interact [102]. Although they are thought to express low levels of several connexins, SGCs and astrocytes primarily form GJs out of connexin 43 (Cx43, gene name *Gja1*), a protein which has garnered attention as a potential therapeutic target [103]. Connexins can also function as monomers, serving not as GJs, but as pores for paracrine and autocrine signaling; these pores are termed hemichannels [104].

Gap junctions connect many of the satellite cells within neuron–glia units to one another [34]. In naïve animals, gap junctions are thought to rarely exist between SGCs of neighboring neurons, although these cells can still influence one another through paracrine signaling. Following nerve injury, more GJs form in the DRG, both within neuron–glia units and between them [105,106]. Injection of Lucifer dye into the DRG of axotomized rats revealed that coupling between SGCs of the same neuron became more than 50% more common, and that the incidence of coupling between neuron–glia units increased more than five-fold [107]. A similar trend has been identified in rats with stenosis and in mice with CIPN [15,71,108]. This increased connectivity can cause DRG neurons to become indirectly “coupled” in models of inflammatory and neuropathic pain, such that they no longer fire totally independently [38]. Interestingly, when TG of naïve rats were treated with siRNA to *Cx43*, an increase in pain-associated behavior was observed [85,109], implying a role of SGC connectivity in physiological pain homeostasis. In contrast, rats with neuropathic facial pain exhibited nearly a week of normalized mechanical sensitivity after the silencing of *Cx43* mRNA [109]. Supporting this, SGC *Cx43* was found to be upregulated via scRNAseq following nerve injury [74]. There are conflicting reports about the effects of age on SGC GJs; one study found older mice to express lower levels of *Cx43* and CX43 in DRG than younger mice [110], while other studies found that SGC dye coupling was increased in aged mouse DRGs, indicating more widespread GJs [111,112].

Astrocytes form numerous gap junctions with one another, contacting countless cells through their extensive network of processes. In one report, sciatic nerve constriction upregulated Cx43 in the DRG, but not the spinal cord, at one week post-injury [38]. This could perhaps by explained by SGCs becoming activated sooner than astrocytes; other studies have identified prolonged upregulation of astrocytic Cx43, particularly in the ipsilateral dorsal horn [82]. A portion of these GJs may form with microglia [113]. Similar to its effect in the DRG, downregulation of spinal cord Cx43 in naïve mice via intrathecal siRNA could cause mechanical allodynia, in part through IL-6 [114]. In mice with a peripheral nerve injury, intrathecal carbenoxolone, a potent decoupler of gap junctions, can transiently reverse mechanical allodynia. Interestingly, one report found that contralateral, but not ipsilateral, mechanical allodynia was affected [115], while others reported bilateral efficacy [116]. Astrocytic GJs likely promote nerve injury-induced neuroinflammation by responding to and further driving cytokine release [116]. Consistent with a vicious cycle of GJ-enabled activation, inhibition of Cx43 leads not only to reduced mechanical hypersensitivity, but also to reduced Cx43 expression [117]. While astrocytes also express Cx30, in a model of spinal cord injury using transgenic mice, Cx43 was necessary for the establishment of neuropathic pain, while Cx30 was not [118]. Mice lacking both channels also had lower levels of GFAP upregulation than controls [118].

### 3.4. Pannexins

Pannexins, homologues of connexins, are membrane proteins which form channels through which purines can escape [119]. They do not form GJs, but instead function as pores, allowing for autocrine and paracrine signaling. Panx1 expressed in sensory neurons, SGCs, microglia, and hematopoietic cells has been separately shown to be important for pain signaling [120,121,122,123]. Blocking Panx1 in cultured DRG neurons reduces P2 × 3-dependent calcium influx [124] and P2X7-dependent influx into SGCs [125]. Current studies disagree about whether Panx1 upregulation primarily occurs in sensory neurons or their satellite cells, with one study finding expression mainly in neurons following SNI surgery in rats [126], and another finding that deletion of Panx1 in GFAP-expressing cells greatly reduced the pain response in a murine model of orofacial inflammatory pain [123]. Under homeostatic conditions, single-cell RNAseq identified higher and more widespread *Panx1* expression in TG neurons than in supporting glia [127]. Astrocytes also express Panx1, which plays a critical role in ATP signaling, can bind to P2X7, and can potentiate the release of IL-1β [128,129,130]. Surprisingly, Panx1 does not appear to be upregulated in the murine SC at 14 days after SNI, and astrocytic Panx1 has yet to be implicated in neuropathic pain [126].

### 3.5. ATP Signaling

ATP, once thought only to serve as a source of energy, is now recognized as a primary means of communications between neurons and glia. Two families of purine receptors exist; P2X receptors are fast-acting, purine-gated cation channels, while the P2Y family are GPCRs [131]. P2X7 knockout mice are resistant to inflammatory- and neuropathic injury-induced hypersensitivity [132]. Systemic inhibition of P2X7 receptors via I.P. injection of a BBB-permeable small molecule could dramatically reduce hypersensitivity after CFA injection into the hindpaw and after nerve injury; this effect was lost in IL-1-alphabeta knockout mice [133,134]. A similar study identified P2X1, P2X7, and P2Y12 inhibition as capable of transiently reversing mechanical allodynia [135]. Although glia likely contribute to some or all of these effects, which glia are responsible remains unclear.

In the sensory ganglia, nerve injury causes an increase in bidirectional ATP signaling between SGCs and neurons [136,137]. Sensory neurons may initiate this signaling through rapid action potential firing [137,138]. SGCs most prominently express the purine-activated channel P2X7 [139,140]. Under neuropathic pain conditions, SGCs are more sensitive to ATP and express higher levels of P2X7 [141,142]. DRG and peripheral nerve samples from humans with neuropathic pain had enriched P2X7 in both satellite glia and Schwann cells [132]. Activation of P2X7 elicited TNF-α secretion from SGCs [143], while inhibiting it reduced their expression of TNF-α and p-ERK [144]. Interestingly, longer-term activation of SGC P2X7 has been shown to decrease the neighboring sensory neuron’s expression of P2X3 and ultimately desensitize that neuron to painful stimuli [145]. SGCs also express other P2Rs, including P2Y12. Linguinal nerve injury caused P2Y12 upregulation in TG SGCs, and inhibition of P2Y12 alleviated mechanical and thermal sensitivity and reduced GFAP expression in SGCs [146].

Primary sensory neurons express all P2X receptors except P2X7 and are notably enriched with P2X3, which is highly expressed by small- and medium-diameter sensory neurons [140]. P2X2 and P2X3 can become upregulated in chronic pain conditions [137,147], and repeated stimulation of DRG neurons further upregulates P2X3 via CaMKII signaling [148]. This upregulation causes sensitivity to purines; ATP treatment is sufficient to drive ectopic firing in DRG neurons from spinal nerve ligation rats [149]. DRG neurons also express a range of P2Y receptors [150,151].

Although canonical astrocytes express P2Rs and are frequently involved in purine signaling [152], they do not express P2X7 [153]. Instead, P2X7 is highly expressed by spinal microglia, which upregulate it following nerve injury in a pro-nociceptive manner [154,155]. Genetic studies have indicated that mice and humans with mutations preventing P2X7 pore formation are resistant to chronic pain, and that blocking only this pore function is sufficient to provide pain relief [156]. Therefore, although it is accepted that activation of SGC P2X7 drives acute pain and that microglial P2X7 is a therapeutic target of interest for neuropathic pain, the role of SGC-expressed P2X7 in neuropathic pain is unclear.

### 3.6. Cytokine and Chemokine Mediated Immune Signaling

Cytokines are small proteins which regulate immune function through autocrine, paracrine, and endocrine signaling [157]. After nerve injury, both neurons and glia release pro-inflammatory cytokines, contributing to a mutual activation likely involved in the development of chronic pain [158].

Injured SGCs increase their production and excretion of the pro-inflammatory cytokines IL-1β [159], TNF-α [160], and IL-6 [161]. CFA injection into the whisker pad causes TG SGCs to upregulate IL-1β and TG neurons to increase their expression of IL-1R1 [159]. TG neurons from rats with inflammatory facial pain, induced by CFA injection, were more susceptible to IL-1β’s sensitizing effects than neurons from naïve rats [162]. IL-1β can directly regulate sodium channel activities to increase the excitability of sensory neurons in vitro [163]. Nerve injury causes broad upregulation of TNF-α in rat DRG, including in SGCs, while neurons also upregulate TNF receptor 1 (TNFR1, also known as p55) [160,164]. TNF-α is sufficient to enhance sensory neuron sensitivity [165], and blocking TNF-α immediately following injury greatly improves mechanical and thermal hypersensitivity [160]. Likewise, the SGCs in ipsilateral DRG after chronic constriction injury upregulate IL-6 and STAT3 [161]. Neuron-to-SGC paracrine signaling through bradykinin and CGRP has been shown to upregulate P2R levels in SGCs [166,167] and to induce inflammatory cytokine upregulation [168].

Some evidence suggests that at least a subset of SGCs may have further immune-related properties. Early histological examination of SGCs found them to express constitutive MHC class II and to upregulate MHC II after nerve injury [169], an idea supported by recent flow cytometry evidence [88] as well as the identification of immune-like SGCs via single-cell RNAseq clustering [42]. Still, the extent to which these cells engage in functions such as phagocytosis and antigen presentation remains unknown [34].

The post-nerve injury cytokine profile of astrocytes resembles that of SGCs, but implicates additional cytokines. Following nerve injury, astrocytes upregulate TNF-alpha and dorsal horn neurons upregulate the TNF-alpha receptor p55; these effects are most pronounced ipsilaterally [164]. Stimulation of Lamina 1 neurons causes the release of both TNF-α and IL-1β, likely causing effects on other glia as well as dorsal horn neurons [98]. In the later phases of neuropathic pain, astrocyte activation is maintained in part by continued IL-1β signaling facilitated by MMP-2 [170]. In addition, nerve injury-induced upregulation of MMP-2 in DRG SGCs may also contribute to sustained neuropathic pain [170]. Overexpression of TNF-α in SC GFAP+ cells following spinal nerve transection further worsened mechanical allodynia [171].

Several lines of evidence also pointed to an activate role of astrocytic chemokine signaling in neuropathic pain [55]. Brief in vitro exposure to TNF-α caused astrocytes to produce and release CCL2, also called MCP-1 (monocyte chemoattractant protein 1), via JNK (c-Jun N-terminal kinase), which is upregulated in astrocytes after nerve injury [172]. Furthermore, following adoptive transfer of TNF-activated astrocytes via intrathecal administration causes nociception, at least in part through the release of CCL2 by astrocytes [173]. Notably, CX-43 activation is responsible for the release of several chemokines (e.g., CCL2 and CXCL1) from astrocytes that can enhance nociceptive synaptic transmission and neuropathic pain [116]. Additionally, astrocytes also express the chemokine receptor CCR5, which can respond to neuron-produced CXCL13 by driving astrocyte activation and neuropathic pain [174].

### 3.7. Glutamate Transporter Signaling

Glutamate, an amino acid and excitatory neurotransmitter, is tightly regulated by glial cells in both the peripheral and central nervous systems. Canonically, astrocytes uptake glutamate from the synapse and process it to glutamine, which is then released back to the extra-cellular fluid and often taken up again by neurons, at which point it is commonly converted back into glutamate and re-packaged [175]. GLAST and GLT-1 are glutamate transporters, and glutamine synthetase (GS) catalyzes the conversion from glutamate to glutamine; all three proteins are preferentially expressed by astrocytes in the CNS [175]. Interestingly, despite the absence of synapses, SGCs also express high levels of GS, and immunostaining reveals that they also express GLT-1 and GLAST [176,177].

There is little evidence to suggest that SGCs alter their expression of glutamate-related genes following nerve injury, but this is not the case for astrocytes. Following nerve injury, astrocytes appear to transiently upregulate, then chronically downregulate, GLAST and GLT-1 [56,178,179]. When glutamate uptake is impaired, this glutamate lingers in the dorsal horn and drives excessive activation of dorsal horn neurons via NMDA receptors, causing spontaneous nociception [180]. Downregulation of GLT-1 could be reversed by inhibition of N-myc downstream-regulated gene 2 (NDRG2), alleviating allodynia and hyperalgesia [181].

Nitric oxide (NO) has been proposed to serve as an early and potent driver of SGC activation following nerve injury [61]. Incubation of murine DRG and TG with an NO donor or with capsaicin drove GFAP upregulation, while capsaicin plus NO inhibitor did not [182]. Unfortunately, the immediacy with which this occurs after injury and the sub-second half-life of NO in vivo makes this mechanism difficult to investigate and even more difficult to target therapeutically.

### 3.8. Potassium Channels

Intracellular and extracellular potassium concentrations regulate neuronal sensitivity, and thus are important in pain signaling. Potassium channels can be divided into voltage-gated (K_v_) and inwardly rectifying (K_ir_) [183]. Astrocytes and SGCs express a number of potassium channels and play critical roles in regulating extracellular potassium concentrations [34,49]. Chief among these channels is K_ir_4.1 (aka Kcnj10). K_ir_s are structured such that potassium ions can more easily enter the cell through them than exit. Thus, they have a stabilizing effect on the resting membrane potential (RMP) of the cell [184]. Due in part to this high expression of inwardly rectifying potassium channels, the membrane potential of SGCs and astrocytes is very similar to E_k_ (the potassium equilibrium constant) and more negative than that of neurons.

Downregulation of Kir4.1 in SGCs is a hallmark of murine nerve injury and neuropathic pain, having been identified in a range of models [15,72,185]. Without nearby glia serving as a sponge for potassium ions, neurons are exposed to greater extracellular potassium levels, increasing their excitability. In a model of spinal cord stenosis, Kir4.1 was correlated not only with decreased inwardly rectifying currents in SGCs, but also with significantly increased sensory neuron RMPs [15], an effect also seen in a model of inflammatory facial pain [186]. Further, axotomy of the infraorbital nerve raised the resting membrane potential of TG neurons by more than 5 mV, lowered that of SGCs, and was accompanied by a doubling of spontaneous potential oscillations [107]. Conversely, silencing *Kcnj10* mRNA in the TG of otherwise naïve rats is sufficient to induce mechanical hypersensitivity and signs of spontaneous facial pain [185]. Thus, SGCs control peripheral sensitization through potassium regulation. While Kir4.1 (aka KCNJ10) is the primary K_ir_ in rodents [187], KCNJ3 likely plays a larger role in human SGCs based on mRNA expression data [27].

Murine astrocytes also rely on Kir4.1 as their primary inwardly rectifying potassium channel [188]. Conditional knockout of this potassium channel in astrocytes causes neuronal hyperexcitability, a marked reduction in potassium uptake by brain astrocytes, seizures, and even death 2–4 weeks after birth. Additionally, glutamate uptake was impaired in mice with conditional knockout of Kir4.1 in astrocytes [189].

### 3.9. Fatty Acid Signaling and Metabolism

Recent single-cell and single-nucleus RNAseq data have identified that *Fabp7* and *ApoE* are among the best markers of DRG-resident SGCs, suggesting that these cells may be heavily involved in fatty acid signaling [52,74,190] (Figure 5A). There is currently little evidence to suggest that SGCs alter their expression of FABP7 after nerve injury [27,74]. Astrocytes express lower levels of FABP7 and have been seen to upregulate FABP7 when acutely injured [191]. Apolipoprotein E (Apo-E) is a protein involved in the transport and metabolism of fats in the bodies of mammals, and has been implicated in Alzheimer’s disease [192]. *Apoe* is highly upregulated in spinal cord microglia following nerve injury, and has been implicated in chronic pain pathogenesis [193]. *Apoe* is also upregulated in SGCs post-nerve injury, and may be involved in lipoprotein-mediated SGC–neuron communication [74].

Perhaps inspired by these findings, other studies have begun to investigate other fatty acid signaling pathways in SGCs. In a mouse model of collagen-induced rheumatoid arthritis, mice were found to upregulate autaxin in DRG tissues. Autaxin is a critical enzyme in the synthesis of lysophosphatidic acid (LPA), and its principal receptor, LPA1, is expressed primarily by SGCs (Figure 5B). Furthermore, LPA was sufficient to drive GFAP upregulation in SGCs, supporting a fatty acid-based feed-forward mechanism of peripheral sensitization [194]. Hydroxymethylglutaryl-CoA synthases (HMGCS) are important enzymes in the production of cholesterol and in ketogenesis. Two forms, HMGCS 1 and 2, are both highly expressed by SGCs, and HMGCS 1 is downregulated in models of sciatic nerve injury [27,195]. More broadly, bulk RNA-seq in GS-positive DRG cells identified that cholesterol biosynthesis is downregulated following SNI [88]. Whether and how this change impacts neuronal health and excitability remains to be seen. Finally, peroxisome proliferator-activated receptor alpha (PPARα) is highly expressed in SGCs, and SGG expression of this protein is important for nerve regeneration, a role not previously attributed to SGCs [74].

## 4. Conclusions and Future Directions

Astrocytes and satellite glia play important and similar roles in the induction and maintenance of neuropathic pain. However, critical differences exist which become relevant when devising therapeutic interventions. In conditions of neuropathic pain, both cell types upregulate GFAP, although astrocytes have more been convincingly demonstrated to undergo hypertrophy and proliferation. Both glia consistently engage in bidirectional pro-inflammatory signaling through cytokines such as IL-6, TNFα, and IL-1β, and astrocytes also release CCL2/CXCL1 and TSP4. Gap junction upregulation, and particularly increased Cx43 function, is another shared feature. SGCs more reliably downregulate the potassium channel Kir4.1 and increase their expression of the ATP-gated cation channel P2X7. Upregulated signaling through pannexin pores is better characterized in the sensory ganglia than in the CNS, although exactly which cells are involved is still unclear. Astrocytes have been shown to alter their expression of glutamate transporters, a phenomenon not well-characterized in SGCs. As the field learns more about these glia, it is likely that more differences will emerge and further challenge the notion that SGCs are “peripheral astrocytes”. Given the location of SGCs in the PNS, these cells can be readily targeted with CNS-impermeable drugs, including biologics (e.g., monoclonal antibodies), thereby providing more options and avoiding the side effects of many CNS drugs (e.g., mood changes, anxiety, dizziness, sleep disturbance). However, it is possible that successfully targeting astrocytes in the CNS may offer greater leverage, as the critical role of astrocytes in the late phase of neuropathic pain is very well-supported.

## Figures and Tables

**Figure 1 cells-12-00965-f001:**
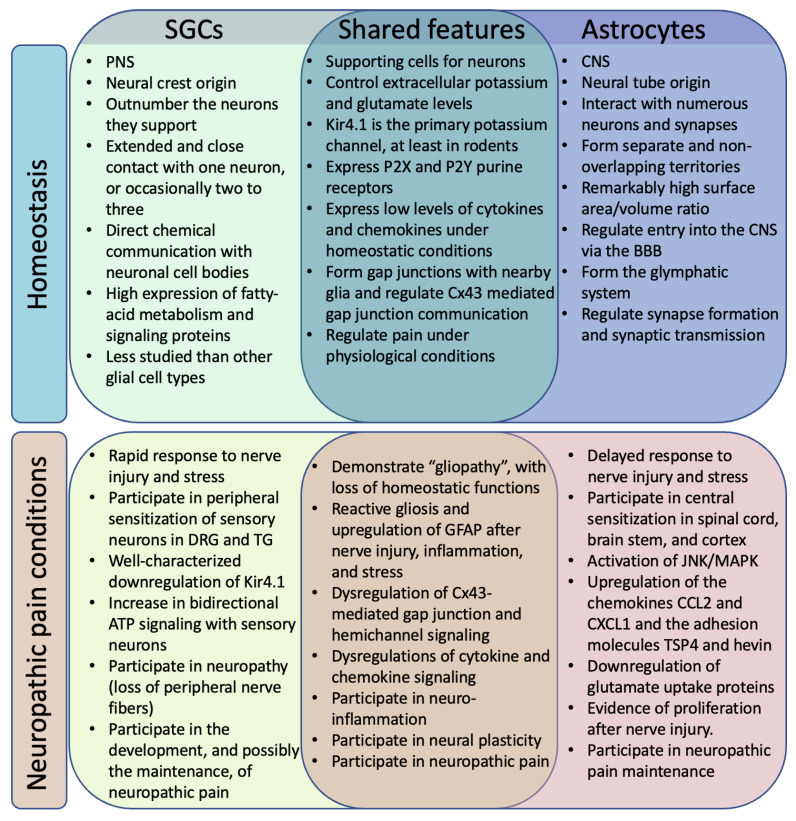
Similar and distinct roles of SGCs and astrocytes in homeostasis and neuropathic pain conditions.

**Figure 2 cells-12-00965-f002:**
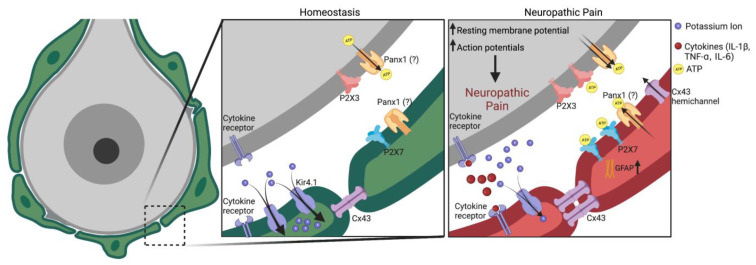
Schematic of SGC roles in homeostasis and neuropathic pain. (**Left**), SGCs surround a sensory neuron and maintain the homeostasis of the extracellular environment of the neuron. (**Middle**), enlarged box of left panel. (**Right**), peripheral nerve injury induces rapid reaction of SGCs in DRG and TG. These reactive SGCs drive neuropathic pain via secretion of neuromodulators, including ATP, cytokines (TNF-α, IL-1β), and chemokines. Nerve injury also results in downregulation of Kir4.1, leading to increases in extracellular K+ levels and neuronal excitability. Neuronal excitability is further enhanced by ATP, cytokines, and chemokine signaling via neuron–glial interactions. Additionally, upregulation of Cx43-mediated gap junction communication after nerve injury may further increase the release of cytokines and chemokines. Figure made with BioRender.

**Figure 3 cells-12-00965-f003:**
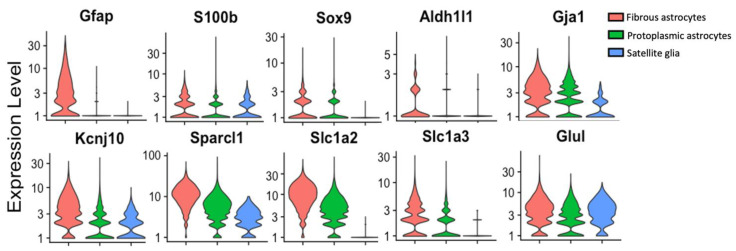
Single-cell analysis showing the expression of cellular marker genes in fibrous astrocytes, protoplasmic astrocytes, and SGCs. *Gfap* (encoding GFAP), *S100b* (encoding S100B), *Sox9* (encoding Sox9), *Aldh1l1* (encoding aldehyde dehydrogenase 1 family member L1), *Gja1* (encoding connexin 43), *Kcnj10* (encoding K_ir_4.1), *Sparcl1* (encoding SPARCL1/high endothelial venule protein, hevin), *Slc1a2* (encoding glutamine-transporter-1, GLT-1), *Slc1a3* (encoding glutamine aspartate transporter 1, GLAST-1), and *Glul* (encoding glutamate-ammonia ligase/glutamine synthetase, GS). Single-cell RNAseq data were accessed from the mousebrain.org database [44] and are presented using Seurat v4.3.0 [53].

**Figure 4 cells-12-00965-f004:**
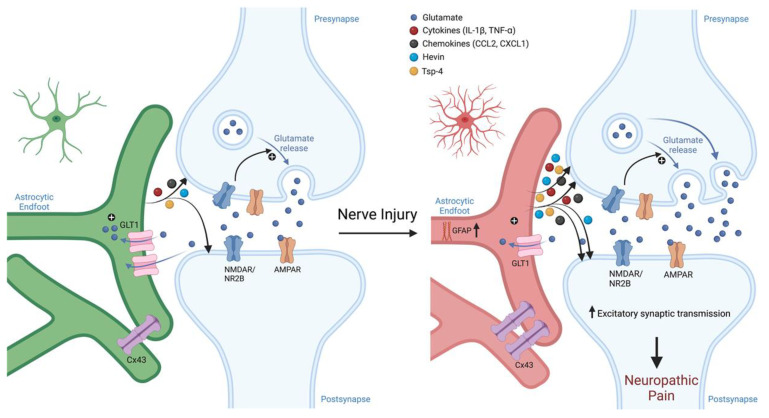
Schematic of astroglial roles in synaptic transmission and neuropathic pain. Nerve injury induces sustained reaction of spinal cord astrocytes. These reactive astrocytes drive neuropathic pain via secretion of neuromodulators, including adhesion molecules (hevin and TSP-4) and chemokines/cytokines. These astroglia-produced neuromodulators can increase the function of NMDA and AMPA receptors at both pre-synaptic and post-synaptic sites, leasing to enhanced excitatory synaptic transmission, central sensitization, and neuropathic pain. Astroglia-produced neuromodulators can further modulate inhibitory synaptic transmission, leading to disinhibition that further exacerbates neuropathic pain. Figure made with BioRender.

**Figure 5 cells-12-00965-f005:**
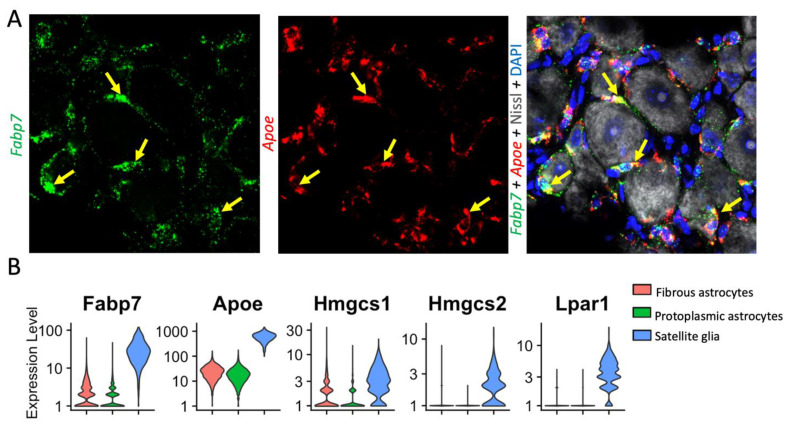
Distinct expression of lipid signaling molecules in SGCs and astrocytes. (**A**) RNAscope in situ hybridization shows *Fabp7* (green, left) and *Apoe* (red, middle) mRNA expression in SGCs (labelled with yellow arrows) of mouse DRG. Right, counter staining of the same DRG section with Nissl (neuronal marker) and DAPI (nuclear marker). Note that Fabp7 and ApoE are not expressed in neurons. To generate Figure 5A, mice were transcardially perfused with 4% PFA, then DRG were further fixed overnight in 4% PFA, dehydrated in sucrose, frozen in OCT, sectioned onto slides, and stained via RNAscope according to the manufacturer’s instructions (Advanced Cell Diagnostics). Neurotrace/Nissl staining was performed thereafter (ThermoFisher). Imaging was performed using a Zeiss 880 inverted confocal microscope. (**B**) Single-cell analysis showing higher expression of Fabp7, ApoE, Hmgcs1 and 2, and Lpar1 in SGCs than astrocytes. *Fabp7* (encoding fatty acid-binding protein 7, aka brain lipid-binding protein 1), *Apoe* (encoding apolipoprotein-E), *Hmgcs1* (encoding HMG-CoA synthetase 1), and *Lpari* (encoding lysophosphatidic acid receptor 1). Single-cell RNAseq data were accessed from the mousebrain.org database [44] and presented using Seurat v4.3.0 [53].

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
