# Peer review of "The Similar and Distinct Roles of Satellite Glial Cells and Spinal Astrocytes in Neuropathic Pain"

_cells, 2023, doi:10.3390/cells12060965_

Round 1
Reviewer 1 Report
This is a well written review article, summarized characteristics and functions of two important types of cells, astrocytes and satellites (SGCs) for their potential involvement in neuropathic pain. The authors comprehensively compare astrocytes and SGCs in homeostasis and with nerve injury, from molecular signaling patterns to structural differences. Widely covered knowledge in this review is very informative. It helps the readers to catch up with the most updated understanding of these cells in neuropathic pain.
Few comments and suggestions:
-As the manuscript is talking about neuropathic pain and the authors state they focus solely on spinal astrocytes, would it be better to add “Spinal” in the title to avoid misleading. “The Similar and Distinct Roles of Satellite Glial Cells and SPINAL Astrocytes in Neuropathic Pain”
-All graphs in the manuscripts are generally very well illustrated. But have a suggestion for SGCs in Fig 2. From our microscope observation, in most cases, SGC ring surrounding sensory neuron is composed of more than 3 cells, usually 6-8 SGCs, and they are roundish in shape, not really flat as endothelial cells in capillaries.
-GFAP is a structure protein within astrocytes and SGCs. Following injury, the protein is increased in these two types of cells. What is the functional meaning of knock out and anti-sense blockade of this protein in the context of neuropathic pain? May I assume blocking GFAP does not block astrogliosis? Does GFAP protein or astrogliosis contribute to neuropathic pain? Where it stays in the cascade of astrocytes/SGCs activation? Suggest elaborating a bit more on functional significance of this protein.
-The author mentioned there is no selective molecular marker for satellite cells (Fig 3). Wondering whether some lipid molecules as analyzed in Fig 5B, HMGCs and/or Lpar1 which seem to be selective for SGCs. Could these molecules be considered as markers?
-Some typos:
Line 238: upregulation not 7pregulation
Line 255: SGC proliferation could also be injury dependent. Would you want to say “independent”? Please check.
Line 543: HMCGs or Hmgcs (fig 5B)? please check, it’s not matching.
Author Response
Reviewer 1
This is a well written review article, summarized characteristics and functions of two important types of cells, astrocytes and satellites (SGCs) for their potential involvement in neuropathic pain. The authors comprehensively compare astrocytes and SGCs in homeostasis and with nerve injury, from molecular signaling patterns to structural differences. Widely covered knowledge in this review is very informative. It helps the readers to catch up with the most updated understanding of these cells in neuropathic pain.
Response: We thank the reviewer for their thorough review.
Few comments and suggestions:
-As the manuscript is talking about neuropathic pain and the authors state they focus solely on spinal astrocytes, would it be better to add “Spinal” in the title to avoid misleading. “The Similar and Distinct Roles of Satellite Glial Cells and SPINAL Astrocytes in Neuropathic Pain”
Response: We appreciate the suggestion and have modified the title, as suggested.
-All graphs in the manuscripts are generally very well illustrated. But have a suggestion for SGCs in Fig 2. From our microscope observation, in most cases, SGC ring surrounding sensory neuron is composed of more than 3 cells, usually 6-8 SGCs, and they are roundish in shape, not really flat as endothelial cells in capillaries.
Response: We modified Figure 2 by adding additional SGCs and changing the shape of SGCs.
-GFAP is a structure protein within astrocytes and SGCs. Following injury, the protein is increased in these two types of cells. What is the functional meaning of knock out and anti-sense blockade of this protein in the context of neuropathic pain? May I assume blocking GFAP does not block astrogliosis? Does GFAP protein or astrogliosis contribute to neuropathic pain? Where it stays in the cascade of astrocytes/SGCs activation? Suggest elaborating a bit more on functional significance of this protein.
Response: You are right. Although GFAP has been extensively used as a marker for reactive astrocytes and SGCs, its function role in pain and astrogliosis is not fully investigated. Kim et al. showed that intrathecal GFAP antisense oligonucleotide treatment in injured rats with neuropathic pain states reversed injury-induced behavioral hypersensitivity and GFAP upregulation in DRG and spinal cord (Ref 65, Kim et al., Pain, 2009), but we do not know the precise mechanism. Future studies are needed to study whether GFAP increase is sufficient to evoke pain hypersensitivity.
-The author mentioned there is no selective molecular marker for satellite cells (Fig 3). Wondering whether some lipid molecules as analyzed in Fig 5B, HMGCs and/or Lpar1 which seem to be selective for SGCs. Could these molecules be considered as markers?
Response: While these genes are relatively enriched in SGCs as compared to astrocytes, they are also expressed by other cell types. Existing markers such as GFAP are useful in that they restrict cells of interest to SGCs + Astrocytes but fail to differentiate between them. Thus, multiple markers are helpful to define SGCs.
-Some typos:
Line 238: upregulation not 7pregulation.
Response: Corrected.
Line 255: SGC proliferation could also be injury dependent. Would you want to say “independent”?
Response: To avoid confusion, this wording was changed to “dependent on the nature of the injury.
Please check.
Line 543: HMCGs or Hmgcs (fig 5B)? please check, it’s not matching.
Response: Changed from HMCGS to HMGCS, encoded by Hmgcs in mice.
Reviewer 2 Report
This manuscript compares and contrast the roles of satellite glia cells (SCGs) and spinal cord astrocytes in homeostatic and pathological pain conditions. The content is very thorough and the manuscript is well-written. The reviewer has the following minor comments.
- Figure 1 is the summarizing figure for the main content of this manuscript describing the similarities and differences between SCGs and astrocytes. However, the reviewer found it is somewhat hard to read and draw clear comparison and contrast information from this figure. It would be easier to read if sub-headings are used along with bullet points. Subheadings used in the text within sections 2 and 3 could be considered here.
- Since the manuscript focuses on spinal cord astrocytes, this can be reflected in the title of the article.
- The manuscript focuses on role of SGCs and astrocytes in neuropathic pain. Some changes described were identified in inflammatory pain models or in cortex upon other CNS insults. As some of these changes could be reasonably expected in neuropathic pain conditions, please thoroughly check the manuscript and references and make sure findings from non-neuropathic pain conditions are clearly identified and described as such.
- In page 3 row 99-100 “Murine DRG neurons have roughly 4-12 SGCs,…”, please clarify that it is the numbers of SGCs per neuron.
- In page 5 row 155 “…where first order nociceptive first order synapses form,…”, please remove the repeated “first order”.
- In page 9 row 360 “P2 x 3” should be “P2X3”. Please correct.
Author Response
Reviewer 2
This manuscript compares and contrast the roles of satellite glia cells (SCGs) and spinal cord astrocytes in homeostatic and pathological pain conditions. The content is very thorough and the manuscript is well-written. The reviewer has the following minor comments.
Response: We thank the reviewer for their thorough review of this manuscript.
Figure 1 is the summarizing figure for the main content of this manuscript describing the similarities and differences between SCGs and astrocytes. However, the reviewer found it is somewhat hard to read and draw clear comparison and contrast information from this figure. It would be easier to read if sub-headings are used along with bullet points. Subheadings used in the text within sections 2 and 3 could be considered here.
Response: We agree with this helpful comment but found it difficult to modify this figure with subheadings because not every category applies to both cell types, and adding “does not apply” seemed to us to only make the figure more crowded. Instead, we slightly modified Figure 1 by separating the homeostasis and neuropathic boxes spatially to better reflect the text and hopefully make the figure more interpretable.
Since the manuscript focuses on spinal cord astrocytes, this can be reflected in the title of the article.
Response: We modified the title as suggested.
The manuscript focuses on role of SGCs and astrocytes in neuropathic pain. Some changes described were identified in inflammatory pain models or in cortex upon other CNS insults. As some of these changes could be reasonably expected in neuropathic pain conditions, please thoroughly check the manuscript and references and make sure findings from non-neuropathic pain conditions are clearly identified and described as such.
Response: We thank the reviewer for the suggestion. Although we focused this review on neuropathic pain, some non-neuropathic pain conditions involving SGCs are also mentioned. We have checked that inflammatory pain studies are introduced as such.
In page 3 row 99-100 “Murine DRG neurons have roughly 4-12 SGCs,…”, please clarify that it is the numbers of SGCs per neuron.
Response: Yes, we modified to “Murine DRG neurons have roughly 4-12 SGCs each, …”.
In page 5 row 155 “…where first order nociceptive first order synapses form,…”, please remove the repeated “first order”.
Response: Removed as suggested.
In page 9 row 360 “P2 x 3” should be “P2X3”. Please correct.
Response: Corrected.